# Microendoscopic Mini-Hemilaminectomy and Discectomy in Acute Thoracolumbar Disc Extrusion Dogs: A Pilot Study

**DOI:** 10.3390/vetsci8100241

**Published:** 2021-10-18

**Authors:** Hiroaki Kamishina, Yukiko Nakano, Yuta Nozue, Kohei Nakata, Shintaro Kimura, Adam G. Drury, Sadatoshi Maeda

**Affiliations:** 1Joint Department of Veterinary Medicine, Faculty of Applied Biological Sciences, Gifu University, 1-1 Yanagido, Gifu 501-1193, Japan; sadat@gifu-u.ac.jp; 2The United Graduate School of Veterinary Sciences, Gifu University, 1-1 Yanagido, Gifu 501-1193, Japan; shinta_ta_ta@yahoo.co.jp; 3The Animal Medical Center, Gifu University, Gifu 501-1193, Japan; ynakano@gifu-u.ac.jp (Y.N.); ytnz.23@gmail.com (Y.N.); k.nakata.1986@gmail.com (K.N.); 4Virginia-Maryland College of Veterinary Medicine, Virginia Polytechnic Institute and State University, 205 Duck Pond Drive, Blacksburg, VA 24061, USA; druryvet@gmail.com

**Keywords:** endoscope, minimally invasive surgery, intervertebral disc extrusion, dog, microendoscopic laminectomy and discectomy

## Abstract

The objective of this study was to evaluate the clinical outcomes and complications of a microendoscopic laminectomy and discectomy (MED) for acute thoracolumbar intervertebral disc extrusions in dogs. Eleven client-owned dogs with acute thoracolumbar intervertebral disc extrusions were included in this retrospective case-series. Dogs were diagnosed with acute thoracolumbar intervertebral disc extrusions using computed tomography (CT) and magnetic resonance imaging (MRI). MED was performed with an integrated endoscopic system to the affected intervertebral disc. Surgery time, intra-operative complications, causes of conversion to microscopic surgery if necessary, post-operative complications, and neurological status on presentation at discharge, as well as any further evaluations in hospital, and long-term concerns via owner contact, were recorded. Post-operative CT images were obtained to compare the extent of laminectomy performed to the planned region of laminectomy. The fully endoscopic procedure was completed in eight dogs without major complications. Three cases were converted to an open surgery due to difficulty removing extruded disc material and controlling hemorrhage. The clinical outcome was good in all cases and equivalent to previously reported prognoses after open surgery. MED is an effective and safe alternative to conventional open procedures in dogs with acute thoracolumbar intervertebral disc extrusion.

## 1. Introduction

Thoracolumbar spinal cord compression is a common neurological condition in dogs, which most commonly results from acute intervertebral disc extrusion with a high incidence in chondrodystrophic breeds [1]. Surgical treatment is recommended when neurological abnormalities are severe or medical treatment fails to ameliorate the associated clinical signs [1,2,3,4]. The goal of surgical treatment for thoracolumbar intervertebral disc extrusion is decompression of the spinal cord. In veterinary medicine, this procedure is often performed with the aid of surgical loupes or a surgical microscope to improve visualization during surgery.

In humans, laminectomy and discectomy using an operating microscope is the gold standard for the treatment of lumbar disc herniation. In recent decades, minimally invasive surgeries have been introduced in lumbar spinal disorders [5]. Specialized equipment, such as operating microscopes, endoscopes, dedicated retractors, and instruments suitable for a small surgical field, are used in minimally invasive spinal surgeries. Some of the advantages of minimally invasive surgery include smaller skin incision, less soft tissue disruption, decreased blood loss, less post-operative pain, better mobility, and faster recovery [6,7]. A prospective randomized controlled study comparing surgical time and clinical outcome between the standard microsurgical discectomy and a minimally invasive microscopic procedure reported equivalent outcomes for lumbar disc prolapse [8]. Other prospective studies in a series of patients undergoing minimally invasive cervical foraminal decompression demonstrated reduced post-operative pain and blood loss and equivalent clinical outcomes compared to open foraminotomy [9,10] or anterior cervical discectomy and fusion [11].

In veterinary literature, most studies have evaluated a minimally invasive approach to the spine in canine cadavers or clinically normal dogs. For the thoracolumbar spine, endoscope-assisted [12,13] and full-endoscopic procedures have been reported [14,15,16]. A minimally invasive device with video assistance was used in studies with canine cadavers and in 10 clinical cases that underwent ventral slot decompression [17]. A dorsal approach to the lumbosacral region using a minimally invasive expandable retractor was reported in a study with canine cadavers; however, an endoscope for the visualization of the surgical field was not used [18]. Wood et al. [19] reported endoscope-assisted lumbosacral foraminotomy in clinically normal dogs. Currently, there are no studies that describe a minimally invasive approach to the thoracolumbar vertebrae in dogs with clinically relevant spinal cord compression secondary to an acute intervertebral disc extrusion.

In the present study, we performed a microendoscopic mini-hemilaminectomy and discectomy (MED) for acute thoracolumbar intervertebral disc extrusion in 11 affected dogs to describe surgical techniques, intra- and post-operative complications, causes of conversion to an open procedure, and clinical outcomes.

## 2. Materials and Methods

### 2.1. Study Design

The present study was conducted as a retrospective clinical study performed on client-owned dogs at the teaching animal hospital of Gifu University. Medical records of dogs that had surgical treatment of acute thoracolumbar intervertebral disc extrusion using microendoscopic technique were identified and included in the study. Informed consent of the owners was obtained prior to video recording of the gait, imaging studies, and surgery.

### 2.2. Imaging Study

In all dogs, anesthesia was induced by propofol at the dose of 5.0 mg/kg and maintained by sevoflurane and oxygen. The level of sevoflurane was set below 1.3 times the MAC of sevoflurane, monitoring the end-tidal concentration of sevoflurane. Magnetic resonance imaging (MRI) (0.4T, APERTO Eterna, Hitachi Healthcare, Tokyo, Japan; 3.0T, Achieva dStream, Philips, Amsterdam, Netherland) was performed to diagnose thoracolumbar intervertebral disc extrusion. Transverse and sagittal images of T2-weighted, T1-weighted, and fluid-attenuated inversion recovery images of the T3-L3 spinal cord segment were acquired. Computed tomography (CT) (Alexion Advance, Canon Medical System Corporation, Tochigi, Japan) with a slice thickness of 0.5–1.0 mm was performed to obtain pre- and post-operative images of the thoracolumbar vertebrae.

### 2.3. Clinical Information and Follow-Up Evaluation

Clinical information of the cases, such as signalment, locations and numbers of disc extrusion, and medical treatment received, were recorded. Neurological grades at presentation, discharge, first hospital visit after surgery, and last hospital visit after surgery were also recorded. In all cases, a telephone interview was performed in order to assess the current neurological status. Neurological grading of the cases followed the scoring system reported in a recent systematic review [4]. The grading system was as follows: grade 0, normal gait; grade 1, thoracolumbar pain with no neurological deficits; grade 2, ambulatory paraparesis; grade 3, non- ambulatory paraparesis; grade 4, paraplegia with intact deep pain perception in at least one limb; and grade 5, paraplegia with loss of deep pain perception. “Ambulation” was defined as the ability to walk 10 consecutive steps without support as previously reported [20].

### 2.4. Preoperative Planning

The extent of the planned laminectomy window was defined based on the degree of spinal cord compression determined by CT and/or MRI. Mini-hemilaminectomy was performed to remove the accessory process and pedicles of the cranial and caudal vertebrae of the affected intervertebral disc space, and spare the articular processes [21]. The cranial and caudal borders of the planned laminectomy were defined by the location where the spinal cord first deviated medially and/or dorsally due to compression by the herniated disc. Similarly, the ventral and dorsal borders were defined by the dorsoventral extent of the spinal cord compression. These borders were defined using preoperative CT. If CT revealed marginal spinal cord compression, MRI was used to aid determining the location and length of spinal cord compression.

### 2.5. Surgical Procedure and Post-Operative Management

All dogs were anesthetized using our standard protocol and maintained with sevoflurane in oxygen as described in the imaging study. Intra-operative analgesia for all study patients during surgery included constant rate infusion of fentanyl (3–15 µg/kg/hr) and ketamine (0.12–0.6 mg/kg/hr). The dogs were positioned in a sternal recumbency on a vacuum beanbag with the pelvic limbs extended caudally. The surgical area was prepared in the usual fashion. All surgeries were performed with an integrated endoscopic surgical system (EasyGO! 2nd generation; KARL STORZ Endoscopy Japan K.K., Tokyo, Japan). Surgical procedures were performed by one of two neurologists, HK or YN. The location of the affected intervertebral disc space was confirmed by a lateral fluoroscopic view (ARCADIS Orbic, Siemens Healthcare, Erlangen, Germany). A 2–3-cm skin incision was made over the surgical area lateral to the midline with a No. 10 scalpel blade.

A dilation sleeve (outer diameter 5.2 mm) was inserted to the level of the target intervertebral foramina at approximately a 70° angle laterally from the spinous process. The location of the dilation sleeve was confirmed by fluoroscopy. Further dilation sleeves were serially placed over the inserted sleeve in increasing size to widen the port of entry (Figure 1A). A tubular retractor with an outer diameter of 15 mm or 19 mm, depending on the size of the dog, was placed over the dilation sleeves, and positioned over the pedicle. The location of the tubular retractor was verified once more using fluoroscopy. The tubular retractor was attached to an articulating locking arm that was clamped to the surgical table, followed by the removal of all dilation sleeves. The telescope was inserted to the telescope holder of the tubular retractor and the light cable and camera head were connected (Figure 1B). A light source (Cold Light Fountain XENON 300; KARL STORZ Endoscopy Japan K.K., Tokyo, Japan) was connected to provide a view of the surgical field on a monitor.

The remainder of the surgical procedure was performed through the tubular retractor. Epaxial muscles and soft tissues covering the pedicle were removed using spoon forceps, electrocautery, and a periosteal elevator (Figure 1C). The accessory process and pedicle were removed using a high-speed drill with a diamond burr with frequent lavage. After the vertebral canal was entered, epidural fat was visualized, and Kerrison rongeurs were used to complete the mini-hemilaminectomy. Herniated disc material was removed with the use of a disc grasper and microcurette (Figure 1D). At the conclusion of surgery, a palpation hook was inserted into the vertebral canal under the spinal cord to ensure sufficient exposure and decompression of the spinal cord. Intra- and post-operative complications were recorded. Post-operative CT was performed immediately after surgery. In cases where conversion to an open approach was required, post-operative CT imaging was not always performed.

For post-operative management of ambulatory cases, neurological status was monitored. For non-ambulatory dogs, frequent turning was performed. Post-operative analgesia for all study patients during hospitalization included constant rate infusion of fentanyl (5 μg/kg/h) and ketamine (0.4 mg/kg/h). Anti-inflammatory medication or analgesics were not prescribed at discharge. At discharge, we instructed the owners to restrain their animals for 2 weeks and recommend leashed or harnessed walks for 2 to 4 weeks post-surgery if the dogs were ambulatory. Basic physiotherapy, including standing proprioceptive feedback and passive range of motion exercises, was also performed by owners for 2 weeks post-surgery. Frequent turning was instructed if the dogs were non-ambulatory.

### 2.6. Imaging Analyses

Pre- and post-operative CT images were exported in the DICOM (Digital Imaging and Communication in Medicine) format to a commercial imaging software for imaging analyses. The planned craniocaudal and dorsoventral borders of laminectomy were defined as described and the area of laminectomy was obtained by manually tracing the defined borders, using the preoperative CT images (Figure 2A,B). On post-operative CT images, the length and area of laminectomy were measured. The area of laminectomy was measured by manually tracing the border of the bone defect (Figure 2C,D).

## 3. Results

### 3.1. Case Population

Eleven dogs with acute thoracolumbar intervertebral disc extrusions were included in this study. The breeds and numbers of dogs were as follows: Miniature Dachshund (*n* = 5), French Bulldog (*n* = 1), Shih Tzu (*n* = 1), Labrador Retriever (*n* = 1), Cardigan Welsh Corgi (*n* = 1), Boston Terrier (*n* = 1), and mixed breed (*n* = 1). Nine dogs were male, one of which was castrated, and two dogs were female, both of which were spayed. Median body weight was 10.7 kg (range, 4.4–34.8 kg). Median age at surgery was 101.2 months (range, 42–158 months). Ten dogs had a single-level intervertebral disc herniation. One dog had multi-level intervertebral disc extrusion in two locations at L2-3 and L3-4. The mean duration of clinical signs was 15.7 days (range, 2–45 days). Medications prior to surgery included prednisolone, firocoxib, and meloxicam. Ten dogs received one of these mediations prior to surgery. Six dogs (Dog number 1, 4, 6, 8, 9, and 11) were on prednisolone (the dose ranged from 0.5mg/kg q24h in four dogs to 1.0 mg/kg q24h in two dogs). Two dogs (Dog number 2 and 3) received firocoxib (the doses were 3.3 mg/kg q24h and 3.0 mg/kg q24h). The remaining two dogs ((Dog number 5 and 10) received meloxicam (the doses were 0.1 mg/kg q24h and 0.15 mg/kg q24h). Clinical information of all cases is summarized in Table 1. At presentation, neurological deficits in most dogs were considered to be moderate to severe; the median neurological grade was 3 (range, 2–5) (Table 2).

### 3.2. Surgery and Intraoperative Complications

Six dogs received mini-hemilaminectomies with a 15 × 40-mm (diameter, length) working tube and five dogs with a 19 × 70-mm working tube. In eight dogs, mini-hemilaminectomy and discectomy was performed via full endoscopic procedure (Figure 1A–D). In these dogs, no major intraoperative complications were encountered. Conversion to an open surgery was needed in the remaining three dogs. The cause of conversion was difficulty in the removal of herniated disc material (*n* = 3). In one of these dogs, hemorrhage from the internal vertebral venous plexus was also the cause of conversion. The mean surgery time was 148 min (range, 80–237 min) for all dogs and 136 min (range, 80–237 min) for dogs receiving full endoscopic procedure. The mean surgery time was longer for dogs that received conversion to open surgery (mean, 220 min; range, 148–236 min). Data of individual cases are presented in Table 2.

### 3.3. Post-Operative Complications and Clinical Outcomes

None of the dogs had post-operative complications as either a direct or indirect result of surgical procedures. At presentation, neurological deficits in most dogs were considered to be moderate to severe; the median neurological grade was 3 (range, 2–5).On the day of discharge (mean post-operative day 2, range 0–5), the neurological grades of five dogs (Dog number 1, 2, 3, 5, and 8) improved and those of other dogs (Dog number 4, 6, 7, 9, 10, and 11) remained at the same grades to their pre-operative grades. At discharge, 6 dogs (Dog number 1, 2, 3, 9, 10, and 11) were ambulatory and 5 dogs (Dog number 4, 5, 6, 7, and 8) were non-ambulatory. Eight dogs were evaluated within 3 weeks after surgery. At the first evaluation (median post-operative day 15.5, range 14–21) the neurological grades were improved in six dogs (Dog number 3, 4, 5, 6, 9, and 10) from their neurological grades at discharge, however 2 of these dogs (Dog number 4 and 5) remained non-ambulatory. Further post-operative evaluation was performed in three dogs (Dog number 1, 4, and 6) on post-operative days 415 (*n* = 1), 55 (*n* = 1), and 35 (*n* = 1). Dog number 6 further improved its neurological grade (from grade 2 to grade 0). Dog number 4 (grade 3) and Dog number 1 (grade 2) remained in the same grade from the first evaluation. The other dog (Dog number 5) that was non-ambulatory at the first evaluation was lost to follow-up. 

Long-term assessment was performed by telephone interview to the owners. The median post-operative day of telephone interview was 398 days (range, 150–1638). In six dogs that had neurological deficits at their last in-hospital evaluation, the neurological grades were further improved in five dogs and remained at the same grade in one dog. The median neurological grades obtained by telephone interview was 0 (range, 0–2). In five dogs that were neurologically normal (grade 0) at their last evaluation, all of them remained normal at the telephone interview. Data of individual cases are presented in Table 2.

### 3.4. Morphometric Analyses

Post-operative CT images were obtained immediately after surgery in eight cases. The laminectomy dimensions were measured on post-operative CT imaging and the extent of laminectomy was compared to the planned region for eight laminectomies in seven cases that underwent full endoscopic procedure (Figure 2A–D). The mean ± SD ratios of the actual length and area of laminectomy in relation to the planned length and area of laminectomy were 106.6 ± 16.1% and 109.1 ± 23.8%, respectively. There was an equal distribution of the number of cases that had received either smaller or bigger laminectomy than the planned laminectomy size. Data of individual cases are presented in Table 3.

## 4. Discussion

MED using an operating endoscope and a tubular system for lumbar disc herniation in humans was introduced by Foley and Smith in 1997. The technique is regarded as an excellent minimally invasive spine surgery, increasingly replacing conventional open techniques. MED can be advantageous over an open approach because it avoids extensive detachment of paraspinal muscles from the lamina, spinous process, articular process, and pedicle. These features of MED have been shown to reduce the amount of intraoperative bleeding, post-operative pain and scarring, and post-operative infection rates, all favoring early discharge. In veterinary literature, a minimally invasive approach to the thoracolumbar vertebral canal has been reported only in canine cadavers or clinically normal dogs [12,13,14,15,16,18,19,22]. The objective of the present study was to report our initial experience of MED in dogs with thoracolumbar intervertebral disc herniation. The specific aims were to describe the surgical technique and report intra- and post-operative complications, causes of conversion to open surgery, and the clinical outcome in the early adaption of this technique.

The initial neurological grades of the cases at presentation ranged from grade 2 to grade 5. At discharge, the neurological grades of five dogs had improved and those of the other dogs remained at the same grades as on presentation. None of the dogs worsened post-operatively; thus, MED can be performed without neurological worsening during the perioperative period. It is important to convert to open surgery if intra-operative complications are noted in order to avoid post-operative complications and clinical worsening. In eight dogs that received reassessment within 3 weeks post-operatively, seven dogs (87.5%) improved their neurological grades at the first evaluation. In six dogs that underwent a full endoscopic procedure and reassessment within 3 weeks post-operatively, five dogs (83.3%) improved their neurological grades at the first evaluation. The other dog had an unchanged grade (grade 2). At long-term post-operative follow-up, all dogs were ambulatory, and their neurological status had improved from initial presentation. In a systematic review [4], recovery rates after decompression surgery for grades 3 and 4 were reported to be 93% (95% CI: 90–96) and 93% (95% CI: 88–96), respectively. Although all dogs eventually became ambulatory without support, five dogs remained paraparetic or ataxic (grade 2) at long-term post-operative follow-up. These dogs had the most severe spinal cord injuries (either grade 4 or 5) at presentation. The recovery rate of dogs without intact deep pain perception was reported to be 61% [4]. As the long-term post-operative follow-up was conducted by telephone interview, neurological grading may have been inaccurate because dogs were graded as grade 2 unless the owners reported that the dog was walking completely normally. Nonetheless, the clinical outcome of dogs in this study after MED was comparable, if not superior, to open decompression surgeries.

Significant intra-operative complications were only encountered in one dog that experienced uncontrolled hemorrhage from the internal vertebral venous plexus, requiring conversion to an open surgery. Inability to control hemorrhage was partially due to availability of micro-instruments that were suitable for MED in dogs. The instruments, such as nerve hooks, probes, and curettes, designed for human MED are generally oversized for most dogs and as such, development of dedicated instruments for veterinary use is essential. In humans, reported common intra-operative complications include dural tears, nerve root injury, and recurrent herniation. Although excellent illumination and magnification could be achieved by both endoscopic and microscopic surgeries, the diminished perception of depth is one the disadvantages of endoscopic surgery compared to microscopic surgery [23,24,25]. Bi-dimensional vision in endoscopic surgeries is possibly linked to increased risk of iatrogenic dural and nerve root injuries [24,26]. The restricted working space due to the use of a tubular retractor also complicates maneuvers to control hemostasis. The other two dogs that required conversion to open surgery had difficulty removing extruded disc materials during MED. In humans, a lower chance of identifying and removing free fragments of disc materials was associated with the limited ability of the surgeon to orientate the surgical instruments by the restricting confines of the tubular retractor [24,25,26]. However, MED has also been described in humans for its decreased incidence of intra- and post-operative complications, intraoperative blood loss, overall surgery time, post-operative pain, post-operative bedtime, and length of hospital stay. These factors seem to be related to the initial learning curve [27,28].

In this study, we performed a mini-hemilaminectomy for all dogs because the extruded disc material was located either ventrally or ventrolaterally to the spinal cord. In addition, all dogs were presented with an acute history of paraparesis or paraplegia, and the imaging findings were suggestive of Hansen type-1 disc extrusion; thus, justifying performing mini-hemilaminectomy for spinal cord decompression. In a previous study by one of the paper’s coauthors (AD), the size of the laminectomy window after hemilaminectomy using the same endoscopic system was a mean laminectomy length of 13.0 ± 1.5 mm for a 19-mm tubular retractor [14]. In the present study, one dog received laminectomy with the 19-mm tubular retractor and post-operative CT evaluation with a laminectomy length of 15.4 mm, and in the other six dogs, a 15-mm tubular retractor was used and the mean length was 11.9 mm. Although the size of the tubular retractor was determined based on the planned laminectomy size, the bone removal was extended to completely remove extruded disc materials. This can be achieved by repositioning the tube retractor during surgery. The angles of access to the vertebral canal were also different; the angle of access was 30° laterally from the spinous process for the previous hemilaminectomy study [14], whereas the angle of access was about 70° for the mini-hemilaminectomy in the present study. In a mini-hemilaminectomy, a more lateral approach was thought suitable for the direct access to the pedicle and to avoid entrapment of epaxial muscles in the surgical field. Although we did not include dogs with Hansen type-2 disc protrusion in our study, the same approach or nearly 90°-angle approach could be used to access the base of the pedicle and the dorsal part of the vertebral body to perform mini-corpectomy. An investigation of surgical techniques of MED for Hansen type-2 is currently underway. A similar approach was reported for a minimally invasive approach to the thoracolumbar vertebral canal [22]. Another possibility is to perform prophylactic fenestration with an endoscopic system. Fenestration of the distant sites can be performed by swinging the tubular retractor cranially or caudally, or repositioning the system over the target disc space.

Limitations of this study include the retrospective data, small sample size, relatively short observation period, and lack of direct comparison to other surgical techniques. In particular, the long-term follow-up evaluations were based on telephone interview to the dog owners. Although judgment of the ambulation status seemed straightforward because all dogs were either completely normal (grade 0) or ambulatory without owner’s support, neurological grading may have been inaccurate. Further, the benefits of the minimally invasive nature of this technique on clinical outcome should be evaluated in future prospective studies. As in other minimally invasive surgeries, minimally invasive spinal surgery requires intensive training and thus the learning curve significantly affects the rate of peri-operative complications and clinical outcomes [29,30]. The present study was intended to report the initial experience of MED; nonetheless, the results were sufficiently encouraging to continue and expand the application of MED to various forms of intervertebral disc diseases in dogs. In addition, development of dedicated instruments for veterinary patients and gaining proficiency with surgical techniques are expected to further improve the success rate of MED in veterinary patients.

## 5. Conclusions

We herein described the application of MED to the surgical treatment of thoracolumbar disc extrusion in dogs. The dimension of laminectomy via endoscope was equivalent to the planned laminectomy size and the extruded disc materials could be retrieved in most cases. Conversion to an open surgery was due to intra-operative hemorrhage and difficulty removing disc material during surgery. These problems may be overcome by development of dedicated surgical instruments for veterinary use. The long-term clinical outcome for the procedure was considered good and equivalent to previously reported prognoses after open surgeries.

## Figures and Tables

**Figure 1 vetsci-08-00241-f001:**
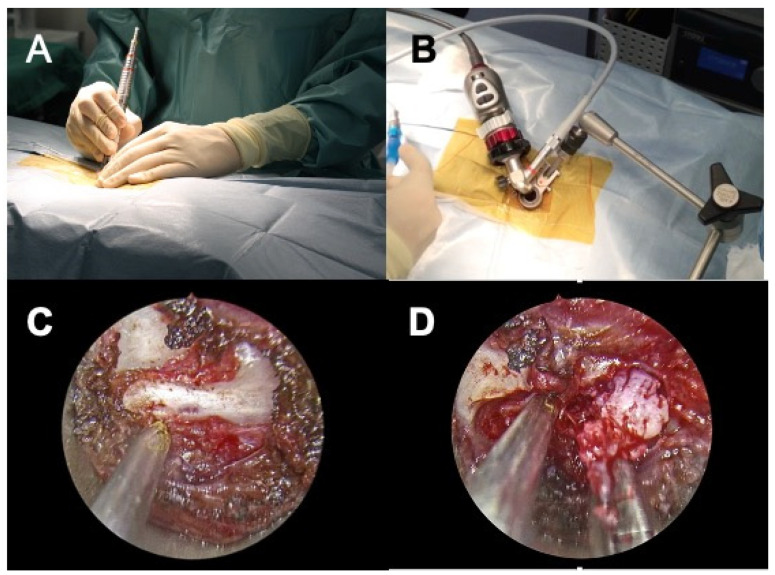
Intra-operative view of microendoscopic laminectomy and discectomy in case No. 9. (**A**), Serial dilation of the muscles is performed by serial insertion of dilation sleeves. (**B**), Complete setup of the system before laminectomy. The camera head and light cable are attached to the tubular retractor that is firmly fixed with the articulating locking arm to the surgical table. (**C**), Intra-operative view after removal of epaxial muscles exposing the accessory process. (**D**), Intra-operative view after mini-hemilaminectomy. The extruded disc materials are removed by a disc grasper (right). The suction tube (left) is placed near the spinal cord to maintain a clear surgical field.

**Figure 2 vetsci-08-00241-f002:**
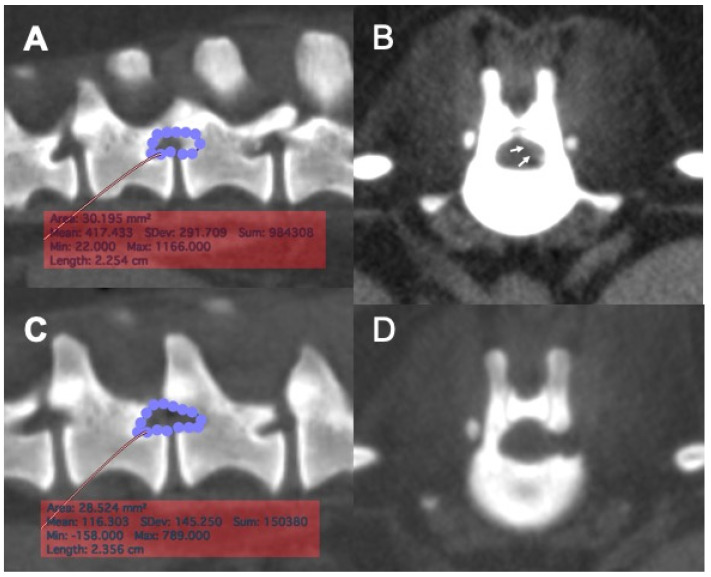
Pre-operative planning and post-operative measurement of laminectomy area with reconstructed CT images. (**A**), The area of the planned laminectomy window was determined based on the location of the extruded disc material in the vertebral canal, using a reconstructed sagittal CT image. The area of planned laminectomy in this case was 30.2 mm^2^. (**B**), A transverse CT image of the intervertebral disc space was used to aid determining the location of the extruded disc material (arrows) in the vertebral canal. (**C**), A post-operative reconstructed sagittal image of the surgical site. The region of laminectomy was traced, and the area of actual laminectomy size was obtained. The actual area of laminectomy in this case was 28.8 mm^2^. (**D**), A post-operative transverse CT image of the surgical site showing the location of mini-hemilaminectomy.

**Table 1 vetsci-08-00241-t001:** Case information of 11 dogs that had microendoscopic mini-hemilaminectomy and discectomy.

Case No.	Breed	Body Weight (kg)	Sex	Age at Surgery (Month)	Clinical Duration (Days)	Locations of IVDH	Prior Medication
1	Cardigan Welsh Corgi	14.6	male	83	2	T12-13	prednisolone
2	Labrador Retriever	34.8	male	121	7	T13-L1	firocoxib
3	Boston Terrier	11.2	male	126	39	T11-12	firocoxib
4	Mix bred	27	castrated male	158	45	T11-12	prednisolone
5	Miniature Dachshund	5.2	spayed female	42	2	T13-L1	meloxicam
6	Shih Tzu	5.8	spayed female	96	17	T12-13	prednisolone
7	Miniature Dachshund	5.8	spayed female	131	6	L2-3, L3-4	none
8	Miniature Dachshund	5.7	male	64	5	L4-5	prednisolone
9	Miniature Dachshund	10.7	male	117	5	T13-L1	prednisolone
10	French Bulldog	13	male	110	23	T11-12	meloxicam
11	Miniature Dachshund	4.4	male	65	22	T11-12	prednisolone

**Table 2 vetsci-08-00241-t002:** Neurological grades of 11 cases at different time points.

Case No.	Neurological Grade (Post-Operative Day)				
	Presentation	Discharge	1st Recheck	Last Recheck	Telephone Interview
1	4	2 (2)	2 (14)	2 (415)	2 (1638)
2	2	0 (5)	-	-	0 (1041)
3	3	2 (2)	0 (14)	-	0 (1064)
4	4	4 (2)	3 (20)	3 (55)	2 (281)
5	5	4 (2)	3 (21)	-	2(512)
6	3	3 (1)	2 (21)	0 (35)	0 (150)
7	4	4 (0)	-	-	2 (398)
			-	-	
8	5	4 (3)	-	-	2 (387)
9	2	2 (1)	0 (15)	-	0 (363)
10	2	2 (2)	0 (16)	-	0 (329)
11	2	2 (1)	2 (15)	-	0 (447)

**Table 3 vetsci-08-00241-t003:** Locations of mini-hemilaminectomy, size of tubular retractor, cause of conversion to an open surgery, and dimensions of planned and actual laminectomy. NA indicates not applicable.

Case No.	Locations of Laminectomy	Size of Tubular Retractor (mm)	Cause of Conversion	Total Surgery Time (min)	Laminectomy Size					
					Planned Area (mm^2^)	Actual Area (mm^2^)	Actual /Planned (%)	Planned Length (mm)	Actual Length (mm)	Actual /Planned (%)
1	T12-13	19 × 70	NA	80	93.7	84.6	90.3	14.6	15.4	105.5
2	T13-L1	19 × 70	residual disc materials hemorrhage	237	NA	NA	-	-	-	-
3	T11-12	19 × 70	residual disc materials	236	NA	NA	-	-	-	-
4	T11-12	19 × 70	residual disc materials	220	NA	NA	-	-	-	-
5	T13-L1	15 × 40	NA	148	41.5	66.8	161.0	12.5	13.6	109.2
6	T12-13	15 × 40	NA	131	37.5	39.6	105.6	9.0	8.5	94.6
7	L2-3	15 × 40	NA	215	26.3	32.1	122.1	7.4	10.5	141.4
	L3-4				47.0	44.0	93.6	12.1	11.7	96.8
8	L4-5	15 × 40	NA	106	69.3	64.5	93.1	16.4	16.2	99.0
9	T13-L1	15 × 40	NA	127	30.2	28.5	94.4	9.7	8.9	91.5
10	T11-12	19 × 70	NA	141	NA	NA	-	-	-	-
11	T11-12	15 × 40	NA	179	44.2	50.0	113.1	12.3	14.1	114.7

## Data Availability

All data sets obtained and analyzed during the experiment are available up on reasonable request from the respective author.

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
