# Peer review of "Microendoscopic Mini-Hemilaminectomy and Discectomy in Acute Thoracolumbar Disc Extrusion Dogs: A Pilot Study"

_vetsci, 2021, doi:10.3390/vetsci8100241_

Round 1
Reviewer 1 Report
I think this is an important contribution, in a clinical area that is badly in need of innovation. It is hard to imagine why, as the techniques are more carefully adapted to veterinary surgery, outcomes should not surpass open techniques. I think the authors would be justified in making even more clear why this technique is likely to produce less inflammation and ultimately produce better outcomes. In addition, would it be possible to use PEG or other medical treatment concurrently with this procedure?
Author Response
Response letter to Reviewer #1
General comments:
I think this is an important contribution, in a clinical area that is badly in need of innovation. It is hard to imagine why, as the techniques are more carefully adapted to veterinary surgery, outcomes should not surpass open techniques. I think the authors would be justified in making even more clear why this technique is likely to produce less inflammation and ultimately produce better outcomes. In addition, would it be possible to use PEG or other medical treatment concurrently with this procedure?
Response:
We are deeply grateful to the reviewer for the constructive comments. We truly agree with the reviewer that the advantages of minimally invasive surgery need to be investigated in the future ideally in a style of prospective randomized control study. We are currently recruiting cases in our prospective study to prove the benefits of endoscopic spinal surgery.
Expected benefits are described in the introduction of the original manuscript, such as smaller skin incision, less soft tissue disruption, decreased blood loss, less post-operative pain, better mobility, and faster recovery.
In the discussion, we stated that the benefits of the minimally invasive nature of this technique on clinical outcome should be evaluated.
We further added to the discussion the need of a future prospective study to compare the clinical outcomes of MED.
The following sentence was added to discussion:
Further, the benefits of the minimally invasive nature of this technique on clinical outcome should be evaluated in future prospective studies.
This change appears in line 392-394.
Regarding the concurrent use of PEG, it is certainly worth trying if such agent is of clinical benefit. The clinical benefits of PEG and MPSS in dogs with acute IVDD were recently evaluated but were not proven in a clinical trial (Olby NJ et al. J Vet Intern Med. 2016 30(1):206-14). At this point, we thought recommendation of concurrent medical therapy needs further study and decided not to include in the revised manuscript.
Reviewer 2 Report
This is an interesting study, about the treatment for acute thoracolumbar intervertebral disc extrusion using a microendoscopic mini-hemilaminectomy and discectomy (MED) in 11 dogs. Currently such minimally invasive approach was evaluated only in canine cadavers or clinically normal dogs, therefore this study has considerable scientific interest. However some limitations need to be discussed before accept it for publication.
The procedure seems to take longer than open mini-hemilaminectomy.
"The mean surgery time was 148 minutes (range, 80-237 minutes) or mean: 220 minutes (range, 148- 236 minutes) for dogs that received conversion to open surgery."
It need to be discussed because a long-lasting anesthesia can be unsafe in dogs with heart, kidney, etc. disorders. Therefore some advice in term of patient selection may be useful.
You suggest that the clinical outcome of dogs in this study after MED was comparable, if not superior, to open decompression surgeries. But looking at the table 3, in the cases 4, 5, 7, 8 and 11 the neurological grades were given by the owner's interview and resulted improved compared to those found at the latest neurological examination. I'm wondering if such interpretations made by the owner could represent a bias and not reflect the true neurological conditions. Please discuss it as limitations of the study.
Finally, several retrospectives and prospective studies support the concept that prophylactic fenestration of the intervertebral disc both at sites of current extrusion and at distant sites, is generally successful in reducing future extrusion of disc material at fenestrated disc spaces. Authors should mention the exclusion of disk fenestration possibility with the MED technique and that It can potentially affect the outcome.
Author Response
Response letter to Reviewer #2
General comments:
This is an interesting study, about the treatment for acute thoracolumbar intervertebral disc extrusion using a microendoscopic mini-hemilaminectomy and discectomy (MED) in 11 dogs. Currently such minimally invasive approach was evaluated only in canine cadavers or clinically normal dogs, therefore this study has considerable scientific interest. However some limitations need to be discussed before accept it for publication.
The procedure seems to take longer than open mini-hemilaminectomy.
"The mean surgery time was 148 minutes (range, 80-237 minutes) or mean: 220 minutes (range, 148- 236 minutes) for dogs that received conversion to open surgery."
It need to be discussed because a long-lasting anesthesia can be unsafe in dogs with heart, kidney, etc. disorders. Therefore some advice in term of patient selection may be useful.
Response:
We thank the reviewer for the constructive and informative comments that significantly improved our revised manuscript.
As described in the original manuscript, the mean surgery time was 136 minutes (range, 80-237 minutes) for dogs receiving full endoscopic procedure. The mean surgery time was longer for dogs that received conversion to open surgery (mean, 220 minutes; range, 148-236 minutes). Therefore, the surgery time was shorter in dogs receiving MED than dogs underwent conversion open surgery. The mean surgery time of 148 minutes was from the surgery time of all 11 dogs.
You suggest that the clinical outcome of dogs in this study after MED was comparable, if not superior, to open decompression surgeries. But looking at the table 3, in the cases 4, 5, 7, 8 and 11 the neurological grades were given by the owner's interview and resulted improved compared to those found at the latest neurological examination. I'm wondering if such interpretations made by the owner could represent a bias and not reflect the true neurological conditions. Please discuss it as limitations of the study.
Response:
We agree to the above comment, the grading might have been biased by the owners. However, the decision seemed straightforward since all dogs were either completely normal or self-ambulatory at the time point. In the revised manuscript, we discussed this limitation as follows:
In particular, the long-term follow-up evaluations were based on telephone interview to the dog owners. Although judgment of the ambulation status seemed straightforward because all dogs were either completely normal (grade 0) or ambulatory without owner’s support, neurological grading may have been inaccurate.
This change appears in line 389-392.
Finally, several retrospectives and prospective studies support the concept that prophylactic fenestration of the intervertebral disc both at sites of current extrusion and at distant sites, is generally successful in reducing future extrusion of disc material at fenestrated disc spaces. Authors should mention the exclusion of disk fenestration possibility with the MED technique and that It can potentially affect the outcome.
Response:
We agree to the reviewer that fenestration can be performed with the endoscopic procedure and it may affect the outcome. We added the possibility of adding prophylactic fenestration in discussion as follows:
Another possibility is to perform prophylactic fenestration with an endoscopic system. Fenestration of the distant sites can be performed by swinging the tubular retractor cranially or caudally, or repositioning the system over the target disc space.
This change appears in line 383-386.
Reviewer 3 Report
General comments
This manuscript has a very inserting theme and is a very well written. Also it should be continued, given the promising results, although the main critic is due to data collection and the results that were obtained whitout comparing to previous results such as the CANSORT-SCI (Olby and Tipold, 2021), regarding ambulation.
Specific comments
Line 2: I suggest to change the name of the article, instead of " Microendoscopic mini-hemilaminectomy and discectomy in dogs with acute thoracolumbar disc extrusions" to Microendoscopic mini-hemilaminectomy and discectomy in acute thoracolumbar disc extrusion dogs: a pilot study".
Line 70: It's missing some patients presentation before study design. The presentation of the 11 dogs can be in form of table such as table 1, but it should be before study design. Also it should be added to this table, the neurological grades at presentation.
Line 76: What about medical video recording? It should be also mentioned that it was with owners consent.
Line 88: The authors mentioned that video recording were made during the process, but final assessment is done with a phone interview without a video recording. Whitout a video recording and only by owners interview it is difficult to have certain regarding the final outcome. I think the last medical record in the last hospital visit has more credibility when talking about final assessment.
Line 94: It should be added in this paragraph, what does ambulation mean to the authors. Authors should consider the bibliographic references of CANSORT-SCI (Olby & Tipold, 2021).
Line 107: The authors mentioned " a standard protocol". What do you mean by that ? And regarding maintenance with isoflurane, it should be mentioned the MAC.
Line 149: It mentioned analgesia in the post-operativo, however what about pain at discharge? Authors should better explain this, if pain was present what medication was prescribed or even if some rehabilitation modalities of pain management were performed (e.g. Standard TENS, Interferential TENS, LASER, etc.) This should be mentioned in the discussion. Also consider to add the evaluation of the hamstring muscle tone. If hypotonic, why not consider funcional electrical stimulation of the sciatic nerve?
Line 152: At discharge home recommendations included leash or harness walks. How was this performed in the non-ambulatory dogs? The phrase should be rewritten, specifying this problem. If dogs were ambulatory at discharge, it is important to mention that post-surgical dogs should not jump or climb stairs in an early stage. Be careful when talking about passive range of motion exercises, these don't promote movement or muscle strength, then are not so used in neurorehabilitation as in orthopedic rehabilitation.
Line 184: The medication prior do surgery included prednisolone, firocoxib and meloxicam. This information should be placed in the material and methods section, adding what dose and for how many days in each dog.
The results section from line 210 is confused, it should be better explained in detailed. Try to specify some results that are included in table 3, such as the number of dogs that stay ambulatory and non-ambulatory at discharge and mention important details, such as that in the 1st recheck 2 of the non-ambulatory remained non-ambualtory, and also that in the 2nd recheck on of these missed the follow-up. So, try to describe case by case evolution. Also it is important for better understanding to use same nomenclature during the manuscript and in tables and not different words (eg. evaluation, recheck).
Line 255: Be careful with percentage presentation, you should mention regarding total population and not the use of sub-groups, so 5 dogs in 11 (45,45%). I understand what you mean, so they to rephrase the sentence.
Line 313: In the limitations section it should be added that this data are retrospective and also the limited informations that is obtained through the telephonic interview.
Also, it should be added in the discussion, for example before limitations or even when talking about surgery (line 267), the comparation with general prognosis after open surgeries, such as described in the CANSORT -SCI (Olby and Tipold, 2021), that species that in case of deep pain perception present, recovery is near 96% and in case of deep pain perception absent, recovery is near 60%.
Author Response
Response letter to Reviewer #3
General comments:
This manuscript has a very inserting theme and is a very well written. Also it should be continued, given the promising results, although the main critic is due to data collection and the results that were obtained whitout comparing to previous results such as the CANSORT-SCI (Olby and Tipold, 2021), regarding ambulation.
Response:
We thank the reviewer for the constructive and informative comments that significantly improved our revised manuscript. The major limitation of this study was its retrospective nature and as pointed out the fact that the final evaluation was based on owner interview. We acknowledge these limitations and tried to make clear in the revised manuscript.
Specific comments:
Line 2: I suggest to change the name of the article, instead of " Microendoscopic mini-hemilaminectomy and discectomy in dogs with acute thoracolumbar disc extrusions" to Microendoscopic mini-hemilaminectomy and discectomy in acute thoracolumbar disc extrusion dogs: a pilot study".
Response:
We changed the title as suggested by the reviewer.
Line 70: It's missing some patients presentation before study design. The presentation of the 11 dogs can be in form of table such as table 1, but it should be before study design. Also it should be added to this table, the neurological grades at presentation.
Response:
Since this was a retrospective study, we think only the inclusion criteria should be described in the materials and methods and information of the recruited cases should be in the result section. However, we agree that the neurological status at presentation should be in the case population paragraph.
We moved the following sentence from 3.3. Post-operative complications and clinical outcomes to 3.1. Case population.
At presentation, neurological deficits in most dogs were considered to be moderate to severe; the median neurological grade was 3 (range, 2-5) (Table 2).
Table 3 was renamed Table 2 in the revised manuscript due to the order of appearance.
This change appears in line 215-217.
Line 76: What about medical video recording? It should be also mentioned that it was with owners consent.
Response:
As pointed out by the reviewer, gait was video recorded for review. We added the following sentence:
Informed consent of the owners was obtained prior to video recording of the gait, imaging studies, and surgery.
This change appears in lines 79-81 and 423-424.
Line 88: The authors mentioned that video recording were made during the process, but final assessment is done with a phone interview without a video recording. Whitout a video recording and only by owners interview it is difficult to have certain regarding the final outcome. I think the last medical record in the last hospital visit has more credibility when talking about final assessment.
Response:
In the original manuscript, we described:
“As the long-term post-operative follow up was conducted by telephone interview, neurological grading may have been inaccurate because dogs were graded as grade 2 unless the owners reported that the dog was walking completely normally.”
Nevertheless, the grades may have been inaccurate. We added the following sentence to discussion:
In particular, the long-term follow-up evaluations were based on telephone interview to the dog owners. Although judgment of the ambulation status seemed straightforward because all dogs were either completely normal (grade 0) or ambulatory without owner’s support, neurological grading may have been inaccurate.
This change appears in line 388-392.
Line 94: It should be added in this paragraph, what does ambulation mean to the authors. Authors should consider the bibliographic references of CANSORT-SCI (Olby & Tipold, 2021).
Response:
To better define “ambulation” we added the following sentence:
We cited the references from the CANSORT-SCI study (Jeffery ND et al. 2020).
“Ambulation” was defined as the ability to walk 10 consecutive steps without support as previously reported.
This change appears in line 104-105.
Line 107: The authors mentioned " a standard protocol". What do you mean by that ? And regarding maintenance with isoflurane, it should be mentioned the MAC.
Response:
I have talked to our anesthesiologist and reflected correct information on the revised manuscript. The maintenance was performed by sevoflurane not isoflurane.
Anesthesia was induced by propofol at the dose of 5.0 mg/kg. We do not routinely measure MAC during surgery and therefore are not able to mention it in the manuscript. The level of isoflurane was set below 1.3 times of the MAC of sevoflurane (2.5), monitoring the end-tidal concentration of sevoflurane.
We changed the sentence as follows:
2.2. Imaging study
In all dogs, anesthesia was induced by propofol at the dose of 5.0 mg/kg and maintained by sevoflurane and oxygen. The level of isoflurane was set below 1.3 times of the MAC of sevoflurane, monitoring the end-tidal concentration of sevoflurane.
This change appears in line 83-85.
2.5. Surgical procedure and post-operative management
All dogs were anesthetized using our standard protocol and maintained with sevoflurane in oxygen as described in the imaging study.
This change appears in line 118-119.
Line 149: It mentioned analgesia in the post-operativo, however what about pain at discharge? Authors should better explain this, if pain was present what medication was prescribed or even if some rehabilitation modalities of pain management were performed (e.g. Standard TENS, Interferential TENS, LASER, etc.) This should be mentioned in the discussion. Also consider to add the evaluation of the hamstring muscle tone. If hypotonic, why not consider funcional electrical stimulation of the sciatic nerve?
Response:
We did not use any anti-inflammatory medication or analgesics at discharge.
Therefore, we changed the sentence as follows:
Anti-inflammatory medication or analgesics were not prescribed at discharge.
This change appears in line 166-167.
In addition, the reviewer suggested some rehabilitation modalities of pain management, but these topics seem beyond the scope of this study and we decided not to include in the discussion.
Line 152: At discharge home recommendations included leash or harness walks. How was this performed in the non-ambulatory dogs? The phrase should be rewritten, specifying this problem. If dogs were ambulatory at discharge, it is important to mention that post-surgical dogs should not jump or climb stairs in an early stage. Be careful when talking about passive range of motion exercises, these don't promote movement or muscle strength, then are not so used in neurorehabilitation as in orthopedic rehabilitation.
Response:
The description about home care was rephrased as follows:
At discharge, we instructed the owner to restrain their animals for 2 weeks and recommend leashed or harnessed walks for 2 to 4 weeks post-surgery if the dogs were ambulatory. Basic physiotherapy, including standing proprioceptive feedback and passive range of motion exercises, was also performed by owners for 2 weeks post-surgery. Frequent turning was instructed if the dogs were non-ambulatory.
This change appears in line 167-169.
Line 184: The medication prior do surgery included prednisolone, firocoxib and meloxicam. This information should be placed in the material and methods section, adding what dose and for how many days in each dog.
Response:
Again, since this was a retrospective study, we think only the inclusion criteria should be described in the materials and methods and information of the recruited cases should be in the result section.
We added detail information on medical treatment as follows:
Six dogs (Dog#1, 4, 6, 8, 9, 11) were on prednisolone (the dose ranged from 0.5mg/kg q24h in 4 dogs to 1.0 mg/kg q24h in 2 dogs). Two dogs (Dog#2 and 3) received firocoxib (the doses were 3.3mg/kg q24h and 3.0mg/kg q24h). The remaining two dogs ((Dog#5 and 10) received meloxicam (the doses were 0.1mg/kg q24h and 0.15mg/kg q24h).
This change appears in line 211-215.
The results section from line 210 is confused, it should be better explained in detailed. Try to specify some results that are included in table 3, such as the number of dogs that stay ambulatory and non-ambulatory at discharge and mention important details, such as that in the 1st recheck 2 of the non-ambulatory remained non-ambualtory, and also that in the 2nd recheck on of these missed the follow-up. So, try to describe case by case evolution. Also it is important for better understanding to use same nomenclature during the manuscript and in tables and not different words (eg. evaluation, recheck).
Response:
We modified this part of the manuscript by specifying the grade changes of each case. We also added the important points suggested by reviewer such as the ambulatory status of the dogs, changes of ambulatory status of the originally non-ambulatory dogs, and the dog that was lost to follow-up.
These changes appear in line 243-255.
The nomenclature was kept consistent with “evaluation” throughout the revised manuscript.
Line 255: Be careful with percentage presentation, you should mention regarding total population and not the use of sub-groups, so 5 dogs in 11 (45,45%). I understand what you mean, so they to rephrase the sentence.
Response:
In the revised manuscript, we state the rate of improvement at the first evaluation among all dogs. Eight dogs received evaluation and 7 of them improved giving the improvement rate of 87.5%. We also state the improvement rate in dogs that received full endoscopic procedure and first evaluation (6 dogs). Five of them improved and the improvement rate is 83.3%.
These changes appear in the revised manuscript as follows:
In 8 dogs that received reassessment within 3 weeks post-operatively, 7 dogs (87.5%) improved their neurological grades at the first evaluation. In six dogs that underwent a full endoscopic procedure and reassessment within 3 weeks post-operatively, 5 dogs (83.3%) improved their neurological grades at the first evaluation.
These changes appear in line 310-311.
Line 313: In the limitations section it should be added that this data are retrospective and also the limited informations that is obtained through the telephonic interview.
We included the retrospective data to the limitation paragraph.
This change appears in 387.
Also, it should be added in the discussion, for example before limitations or even when talking about surgery (line 267), the comparation with general prognosis after open surgeries, such as described in the CANSORT -SCI (Olby and Tipold, 2021), that species that in case of deep pain perception present, recovery is near 96% and in case of deep pain perception absent, recovery is near 60%.
Response:
The recovery rate of deep pain perception present case was mentioned in the original manuscript as follows. The meta-analysis by Langerhuus et al. was cited.
In a systematic review [4], recovery rates after decompression surgery for grades 3 and 4 were 93% (95% CI: 90–96) and 93% (95% CI: 88–96), respectively.
These changes appear in line 315-317.
The reported recovery rate of dogs with deep pain perception absent was added to the revised manuscript as follows:
The recovery rate of dogs without intact deep pain perception was reported to be 61% [4].
These changes appear in line 320-321.
Other changes made in the revised manuscript:
Table 3 was renamed Table 2 in the revised manuscript due to the order of appearance.
The location of Table 2 and 3 were changed.
The original sentence “grade 4, paraplegia with intact deep pain perception in at least one limb non-ambulatory” was changed to “grade 4, paraplegia with intact deep pain perception in at least one limb”.
This change appears in line 99-103.
Round 2
Reviewer 3 Report
I thank for all the changes that were done after my revision, but I would like to point out in line 81 the phrase "...end-tidal concentration of sevoflurano", it was probably a mistake and should be replaced for "...end-tidal concentration of CO2".
Author Response
Thank you for your time and comment to our revised manuscript.
As described in the revised manuscript, the level of sevoflurane was set below 1.3 times the MAC of sevoflurane. This was performed by monitoring the end-tidal concentration of sevoflurane, not the the end-tidal concentration of CO2. So, it was done as described.